# Analysis of the 2007–2018 National Health Interview Survey (NHIS): Examining Neurological Complications among Children with Sickle Cell Disease in the United States

**DOI:** 10.3390/ijerph20126137

**Published:** 2023-06-15

**Authors:** Emmanuel Peprah, Joyce Gyamfi, Justin Tyler Lee, Farha Islam, Jumoke Opeyemi, Siphra Tampubolon, Temitope Ojo, Wanqiu Qiao, Andi Mai, Cong Wang, Dorice Vieira, Shreya Meda, Deborah Adenikinju, Nana Osei-Tutu, Nessa Ryan, Gbenga Ogedegbe

**Affiliations:** 1Global Health Program, Department of Social and Behavioral Sciences, ISEE Lab, NYU School of Global Public Health, 708 Broadway, 4th FL, New York, NY 10003, USA; gyamfj01@nyu.edu (J.G.); jl12376@nyu.edu (J.T.L.); fi354@nyu.edu (F.I.); jumokeopeyemi@gmail.com (J.O.); st3908@nyu.edu (S.T.); to790@nyu.edu (T.O.); dorice.vieira@nyulangone.org (D.V.); sm9223@nyu.edu (S.M.); dba245@nyu.edu (D.A.); noseitutu95@gmail.com (N.O.-T.); ryann01@nyu.edu (N.R.); 2Department of Biostatistics, NYU School of Global Public Health, 708 Broadway, New York, NY 10003, USA; wq2027@nyu.edu (W.Q.); am11115@nyu.edu (A.M.); cw1522@nyu.edu (C.W.); 3NYU Health Sciences Library, 577 First Avenue, New York, NY 10016, USA; 4Institute for Excellence in Health Equity, NYU Langone Health, 180 Madison Avenue, New York, NY 10016, USA; olugbenga.ogedegbe@nyulangone.org

**Keywords:** neurological complications, Black children, African Americans, sickle cell disease, social determinants of health, National Health Interview Survey, maternal education, healthcare utilization, healthcare barriers

## Abstract

This study compared neurological complications among a national sample of United States children with or without sickle cell disease (SCD) and evaluated health status, healthcare and special education utilization patterns, barriers to care, and association of SCD status and demographics/socioeconomic status (SES) on comorbidities and healthcare utilization. Data was acquired from the National Health Interview Survey (NHIS) Sample Child Core questionnaire 2007–2018 dataset that included 133,542 children. An affirmation from the guardian of the child determined the presence of SCD. Regression analysis was used to compare the associations between SCD and demographics/SES on neurological conditions at *p* < 0.05. Furthermore, adjusted odds ratios (AORs) were estimated for having various neurological conditions. Of the 133,481 children included in the NHIS, the mean age was 8.5 years (SD: 0.02) and 215 had SCD. Of the children with SCD, the sample composition included male (*n* = 110), and Black (*n* = 82%). The SCD sample had higher odds of having neuro-developmental conditions (*p* < 0.1). Families of Black children (55% weighted) reported household incomes < 100% of federal poverty level. Black children were more likely to experience longer wait times to see the doctor (AOR, 0.3; CI 0.1–1.1). Compared to children without SCD, those with SCD had a greater chance of seeing a medical specialist within 12 months (AOR 2.3; CI 1.5–3.7). This representative sample of US children with SCD shows higher odds of developing neurological complications, increased healthcare and special education services utilization, with Black children experiencing a disproportionate burden. This creates the urgency to address the health burden for children with SCD by implementing interventions in healthcare and increasing education assistance programs to combat neurocognitive impairments, especially among Black children.

## 1. Introduction

In the United States (U.S.), about 100,000 individuals have been diagnosed with sickle cell disease (SCD), with 40% of these individuals being children and adolescents [1,2,3]. Approximately 2000 babies are born with SCD annually in the US [1,2], with 1 out of every 365 births occurring among US Blacks [4,5]. SCD is a life-threatening erythrocyte disorder caused by a point mutation in the beta chain of the hemoglobin protein that leads to a substitution of valine for glutamic acid at the sixth amino acid of the beta globin gene [2,6]. Neurocognitive impairments, common among SCD patients, result in devastating effects if untreated [4,7]. Neuro-developmental complications of SCD can lead to negative socioeconomic and psychological impacts on both SCD patients and their families [7,8]. Children and adolescents with SCD have a suboptimal health-related quality of life owing to pain/pain crisis [9,10,11,12], anxiety [13], depression [13], neurocognitive impairments, and restrictive daily function [14]. A deficit of auditory and visual processing, coupled with reports of short-term memory loss, shows alterations of pathophysiologic effects among children with SCD [1].

Existing literature suggests that due to neurocognitive impairments, academic challenges are prevalent among children and adolescents with SCD [1,15]. Evidence also indicates that robust academic challenges begin in elementary school and persist until high school and beyond [16], especially among those from low socioeconomic status (family income, education, etc.) [17]. In addition, SCD-related health complications and comorbidities, such as infections, pain crises, acute chest syndrome, and stroke, lead to school absences, thus negatively influencing academic achievement [14,18]. Unfortunately, many educators are unaware of the increased need for special services that children with SCD require [3] due to the social determinants of health (SDoH) and the inconsistent evaluation of these contextual factors for SCD [19,20,21,22]. SDoH can be broadly defined as “the conditions in which people are born, grow, live, work, and age [23]”. Khan and colleagues, examined how SDoH impact persons with SCD; the scoping review highlighted that SDoH including neighborhood and built environment, access to healthcare, socioeconomic status (SES), economic stability, and quality of life, have been explored more than other social determinants, such as education, unemployment, food insecurity, discrimination, and social cohesion [19]. Although several areas of SDoH have been addressed in depth (i.e., quality of life) other areas including impact of health literacy, area of residence, and level and location of care received have not been examined for persons with SCD [19].

Currently, evidence on how neurocognitive impairments resulting from SCD impacts Black children and adolescents at the population level, accounting for socioeconomic status and SDOH among a nationally representative sample of Black children in the U.S. are limited. Boulet et al. (2010) examined the 1997–2005 National Health Interview Survey (NHIS) cross-sectional Sample Child data to describe medical conditions and neurocognitive impairments among Black children aged 0–17 years with SCD [7]. The authors found that children with SCD are at a higher risk of intellectual disability and cognitive impairments; and that the etiology of the neurologic complications associated with SCD is not well understood [7]. We used the 2007–2018 NHIS to describe medical conditions of children with SCD [24]; here we characterize the neurocognitive impairments and health services use among children aged 0–17 years with SCD to expand beyond the 2010 study. Our study aims to extend the knowledge of neurocognitive impairments and their impact among all children reported with SCD including Black children, drawing greater attention to SDoH and other contextual factors impacting this population, utilizing a more recent dataset. Specifically, 2007–2018 NHIS data were used to assess neurocognitive status among all children aged 0–17 years diagnosed with SCD compared to children without SCD, with a subgroup analysis conducted for Black children. We hypothesized that Black children with SCD would have higher levels of neurological complications and healthcare utilization than their counterparts without SCD. In this study, we examined (1) the prevalence of comorbid neurological conditions with SDoH domains including (2) healthcare and special education services access and barriers and (3) relationship between SCD status and demographics/SES on comorbid neurological conditions and healthcare utilization.

## 2. Methods

### 2.1. Study Design

#### Secondary Analysis of NHIS Dataset

We conducted a cross-sectional study with a sample of children aged 0–17 years who participated in the Sample Child Core questionnaire portion of the NHIS survey (2007–2018) [25].The total unweighted sample was133,542 children (68,745 male, 64,797 females).

### 2.2. Data Source

#### NHIS Data

NHIS is a continuing survey that was begun in 1957 that documents information on the amount, distribution, and effects of illness and disability in the United States populations [26]. Moreover, NHIS also assesses the services rendered for or because of such conditions and includes data captured by the Sample Child Core questionnaire [26]. NHIS data were collected via an in-person interview.

### 2.3. Sample Child Core Questionnaire

We conducted a sub-analysis of the NHIS dataset using neurological outcomes based on our findings from analysis of the medical conditions of children with SCD [24]. Similar to our analysis of the NHIS medical conditions data, our sub-analysis of neurological conditions focused on individuals between the ages of 0 to 17 that was randomly selected from each participating household as a subject for the Sample Child Core questionnaire [24]. A caregiver (parent or legal guardian) knowledgeable about that child’s medical/neuro-developmental history and current healthcare needs provided proxy responses for the sampled child [24]. NHIS sampled 133,481 children and caregivers were asked about SCD diagnosis or status. The questionnaire includes items inquiring about the child’s health status, functional limitations, and several SDoH domains including healthcare access. SCD status of the child was determined by an affirmative response from the caregiver (parent/guardian) completing the NHIS questionnaire. SES was determined by household income and insurance [24]. Income level was defined according to the federal poverty level and classified into three categories for subsequent analysis: <100%, 100–200%, and ≥200% of the federal poverty level [24,25]. Medical insurance, another indicator of SES, was classified into three categories: public, private, or other [24].

Overall health conditions were obtained from participants’ medical history questions. Health conditions assessed included functional status (walking, playing, etc.), utilization of special equipment, prescription medication used (minimum of at least three months), health status compared to twelve months ago, and number of days of school absences due to illness/injury [24]. For analysis, health status was classified into three categories: better, worse, and about the same. Duration for missing school due to illness was divided into six categories: none, 1–10 days, 11–20 days, 21–30 days, 31–40 days, and more than forty days, similar to our medical conditions analysis [24]. Healthcare utilization was assessed via questions about accessing medical services, including specialists (e.g., psychiatrists, psychologists, etc.) and included any surgical procedures within the past twelve months. The frequency of annual physician visits was stratified into five categories: none, one to five times, six to nine times, ten to fifteen times, and sixteen or more times. Frequency of annual emergency room (ER) visits was stratified into six categories: none, one time, two to three times, four to five times, six to 16 or more times, and underwent surgical procedure. Finally, healthcare barriers were assessed using the following question, “*Have you delayed getting care for {S.C. name} for any of the following reasons in the PAST 12 MONTHS?*”. An affirmative answer from the caregiver and response was registered and utilized for subsequent analysis [24].

The neuro-developmental conditions evaluated included the following: attention-deficit disorder and attention-deficit/hyperactivity disorder (ADD/ADHD); learning disability; intellectual disability; trouble hearing; trouble seeing; and other developmental delays. Attention-deficit disorder and attention-deficit/hyperactivity disorder (ADD/ADHD) were noted if the respondent answered, ‘yes’ to ‘*Ever told by a health professional sample child had ADHD/ADD*?’. Learning disability was indicated if the participant answered ‘yes’ to ‘*Ever told by a health professional that sample child had an intellectual disability also known as mental retardation?*’ Intellectual disability was identified if the respondent answered ‘yes’ to ‘*Ever told by a health professional that sample child had a learning disability?*’ Trouble hearing was identified if the respondent indicated that they were deaf and had a lot or a little trouble hearing without a hearing aid. Trouble seeing was determined if the respondent stated that the child had any difficulty seeing even when wearing glasses or contact lenses.

### 2.4. Statistical Analyses

Statistical analyses were performed using STATA software (version 16.0) [27]. Frequencies and proportions were obtained for all categorical variables and means and standard deviations for continuous variables. All weighted results were calculated to represent national estimates for children between 0–17 years old.

We evaluated frequencies and weighted percentages of demographic characteristics among children of all races (Appendix A) and Black children (Appendix A) by SCD status. We also examined the associations between SCD status and comorbid neurological conditions while controlling for demographic characteristics in the multivariate logistic regression models for all races (Table 1a,b) and for Black children (Table 2a,b) by SCD status. We utilized a similar approach to characterize medical conditions among children with SCD [24]. We conducted analyses of the various potential interactions between SCD status and demographics impacting the neurological condition/health status/healthcare utilization of children using multivariate logistic regression models (Appendix A and Appendix A) [24]. The *p*-values were deemed significant at *p* < 0.05 for measured main effects with adjusted odds ratios (AORs), and interpreted interaction effects with margin plots. The interaction terms were selected from Appendix A and finally participants missing SCD question responses (<1%) were excluded from the analyses [24].

## 3. Results

### 3.1. Characteristics of Sample

In Appendix A, the NHIS study included 133,481 children with a mean age of 8.5 years (SD: 0.02) across the 12 years, and 215 children had SCD (0.16%). Among the children with SCD, 110 are male, mean age 8.2 years, SD:0.29; racial composition includes Black (81.8%), other races (9.77%), and White (7.7%) (Figure 1). Of the children with SCD, 49.1% are from Southern U.S. Of the children’s families, 53.8% reported a household income <100% of the federal poverty level and were on Medicaid and or had state child health insurance (SCHIP) programs as their primary insurance (Figure 2). Of the Black children subgroup, 55% reported a household income <100% of the federal poverty level; and were on Medicaid/SCHIP (Appendix A).

The period prevalence of SCD among all children was 1.47 per 1000 (SE: 1.4; 95% confidence interval (CI): 1.2–1.8) for the 12-year timeframe of the NHIS data collection. Black children had higher prevalence of SCD (7.83 per 1000; SE: 7.94; CI: 6.3–9.4) and increased burden of neurodevelopmental conditions.

### 3.2. Neuro-Developmental Conditions, Health Status, and Healthcare Utilization

Children with SCD were four times more likely to have an intellectual disability (*p* < 0.018) and two and a half times more likely to experience trouble hearing (*p* < 0.039) (Table 1a). Parents of children with SCD were three times more likely to report worsening health status, while children without SCD were more likely to report the same health status over 12 months (*p* < 0.001) (Table 1b). Positive associations between missed school days and SCD were observed for the entire sample, with most children missing 11–40 school days per year (*p* < 0.05).

The AORs for intellectual disability and trouble hearing were non-significant for the Black children only group (Table 2a). African American children were 2.3 times more likely to have problems with their vision. Children of all races with SCD (28.9%) compared to Black children (29.9%) with SCD had at least two or more annual emergency room visits (*p* < 0.001) and were more likely to have seen a doctor more than five times per year (Table 2b). SCD children were seven times more likely to have limited ability to crawl, walk, run, or play, and three times more likely to take prescription medications for at least three months (*p* ≤ 0.001) within one year.

Black children with SCD were more likely to miss 11–40+ school days because of illness/injury than Black children without SCD (*p* < 0.001). For healthcare access barriers, only one Black child with SCD could not afford prescription medications compared to Black children without SCD (Table 2b).

### 3.3. Association of SCD Status and Demographics/SES with Neurological Comorbidities, and SDoH Outcomes including Healthcare, and Educational Service Utilization

We assessed potential interaction effects between SCD and demographics/SES on neuro-developmental conditions and healthcare utilization patterns among the sampled children (Appendix A and Appendix A).

Considering developmental conditions, the effects between SCD and ADD/ADHD depend on insurance coverage status (*p* < 0.05). The probability of ADD/ADHD is smaller for children without SCD with no insurance coverage, in contrast to the results of children with SCD. We observed that age (11–17 years: *p* < 0.008), maternal education (>high school, *p* < 0.028), and private insurance (*p* < 0.05) had interactions with SCD on learning disabilities. Children without SCD, ages 11–17 years had a higher chance of having a learning disability than children 3–5 years old without SCD, but the opposite result was observed when children had SCD. We also examined that age (3–5: *p* < 0.024, 6–10: *p* < 0.022), region (Midwest: *p* < 0.002, South: *p* < 0.028), public insurance (*p* < 0.036), and private insurance (*p* < 0.011) exhibit statistically significant interactions with SCD for trouble seeing. Public insurance plays a more significant role in the association of SCD and trouble seeing than non-public insurance (Appendix A).

In addition, the relationship between taking prescription medication, seeing a medical specialist for at least three months, and SCD depend on age levels. When we used children who were less than three years as the reference group, children without SCD aged 11–17 years had the highest chance of taking prescription medication (3–5: *p* < 0.040, 11–17: *p* < 0.001) and seeing a medical specialist (6–10: *p* < 0.009). Among children with SCD, those ages of 6–10 years had the highest chance. Male (*p* < 0.010) children without SCD had a more negligible probability of taking prescription medication than females but had a more significant likelihood of taking prescription medication than females when they had SCD (Appendix A).

The number of days of school missed annually because of illness or injury (1–10, 11–20) depends on the age levels interacted with SCD. In most states within the U.S., children aged 3 to 5 years are eligible for preschool (pre-K to kindergarten). Pre-K is considered school age and is a relevant age when children first enter school. Therefore, compared to school age children 3 to 5 years old (reference group), children ages 11 to 17 had statistically significant interaction effects with SCD on 1–10 days of school missed (*p* < 0.048) and 11–20 days of school missed (*p* < 0.012). In terms of healthcare and special education services use, insurance coverage (*p* < 0.026) interacted with SCD, had an impact on children who saw a therapist, and income (100–200: *p* < 0.041) interacted with SCD, influenced children who saw an optometrist (Appendix A).

For the number of visits (1–16+ days per year) to the emergency room, the demographics/SES, including age, maternal education, region, income, and insurance, interactions exist (*p* < 0.05) for children with SCD. Male (*p* < 0.012), household income (>200: *p* < 0.022), and private insurance (*p* < 0.001) had enormous effects that interacted with SCD status (Appendix A).

## 4. Discussion

Using the NHIS, we compared neuro-developmental conditions, healthcare and special education services, and barriers to healthcare among children with and without SCD in the U.S. Children with SCD in the U.S. (who are majority Blacks in the NHIS dataset) have significantly more neuro-developmental conditions, higher healthcare utilization, and experience greater healthcare barriers than their non-SCD counterparts, which is consistent with results from the previous 1997–2005 NHIS analyses [7] and various literature on the SDoH for SCD [19,20,21,22]. Although children with SCD have greater doctor’s office and emergency room visits, we observed that caregivers/parents/guardians of these children reported encountering more barriers to healthcare access compared to non-SCD children. Moreover, prescription medication affordability was a significant financial burden for low-income families of children with SCD. The NHIS study highlights that Black children with SCD reported worsening health over a 12-month period that could be due to access to inadequate care [24]. Public health insurance may limit accessibility to specialized healthcare services posing tremendous challenges to receiving timely care [28]. This may result in children settling for emergency room visits as their sole option for primary care [29]. This situation is not ideal for managing SCD, a chronic condition that requires supportive services.

Children with SCD have a higher risk for neuro-developmental conditions. They are more likely to frequently utilize healthcare and special education services, perhaps due to their comorbid disease status [30]. They are more likely to have learning disabilities, intellectual disabilities, trouble hearing, seeing, and/or other developmental delays compared to children without SCD. Furthermore, these children experience more healthcare barriers and stigmatization [31,32] than children without SCD. Literature has established that many Black children with SCD live in households of lower socioeconomic status with parents with low income [17,30] and are significantly impacted by SDoH due to healthcare inequities [33,34]. SDoH including socioeconomic status can directly affect the timeliness and quality of the children’s healthcare, which is consistent with the trends observed from our analyses. Moreover, studies have documented that SCD patients are hesitant to visit the emergency room/urgent care due to previous negative health experiences and stigmatization [32,35] because of the providers’ implicit biases [36].

Ethnically and minoritized persons, and African American/Black with SCD, have been documented to experience more delays in care than non-minoritized individuals [19,37,38,39]. The association between discrimination, disease-related stigma, and healthcare utilization is a significant SDoH for persons with SCD, which has been understudied [19]. Few studies have reported that perceived discrimination in healthcare systems may be a significant risk factor in chronic diseases, and lead to racial and ethnic disparities [19,37,39] and subsequent poor health outcomes for persons with SCD.

Within the context of SDoH, our analyses highlight that Black children with SCD are more likely to utilize healthcare services than Black children without SCD as expected. The finding that Black children with SCD utilize more healthcare services shows that this group might not be receiving the necessary healthcare services to manage their comorbid conditions because the increased frequency of visitations does not equal better management of SCD, and could be indicative of poor management for this chronic condition [40,41]. As demonstrated in the scoping review by Khan and colleagues, [19] and a recent study that highlights the need to utilize evidence-based strategies for SCD management, including using the NHLBI SCD recommendations, individualized pain protocols, electronic health records, and other care-interventions, specifically geared towards improving provider knowledge and mitigating provider bias [42]. Improved management of SCD patients has occurred through various evidence-based interventions, including healthcare teams with a care manager who may be the only healthcare team member that has broad knowledge of the patient’s experience and is effective in providing better care and improving outcomes, which results in decreased visitations [40]. Moreover, it is concerning that Black children with SCD experience more healthcare barriers than Black children without SCD, despite the existence of evidence-based strategies that, when effectively utilized, can improve outcomes.

Studies have reported that Black children with SCD disproportionately need special education services and other school-based interventions [43]. Our analyses indicate that children with SCD are more likely to have learning disabilities, which may affect the child’s self-esteem. Our main findings demonstrate that children with SCD have higher levels of neurocognitive problems and higher healthcare utilization with more healthcare access barriers than those without SCD, which is consistent with the previous NHIS study [44]. In comparison to the 1997-2005 NHIS study, our study call attention to a significant increase in the adjusted odds ratio for Black children who have trouble seeing, which may significantly impact the child’s ability to learn. Low SES serves as a potential barrier as individuals are unable to access certain healthcare resources or are deterred from receiving care due to costs. Interventional measures are pivotal to addressing the healthcare access barriers due to low SES. These barriers include chronic pain that children diagnosed with SCD often experience, the lack of insurance needed to cover the frequent hospital visits, and the stigma that patients with SCD face.

Findings from this study, our previous study on medical conditions [24], and the Boulet et al. NHIS study [7] implicate that when compared to the general population of children in the U.S., children with SCD are more susceptible to developmental conditions than children without SCD. While there are similarities between the outcomes of interest for the three studies, this study extensively examined the interactive effects of SCD status and demographic/SES on comorbid neurological conditions and healthcare utilization. Another distinction is the usage of different scales applied to certain variables. For example, the Boulet et al. NHIS study [7] and this study observe the adjusted odds ratio for emergency room visits, however the Boulet et al. NHIS study [7] used a dichotomous approach such as “more than one visit to emergency department” or “one or less than one visit to the emergency department”. Our study instead observed the same variable using the following categories: no visits, one visit, 2–3 visits, 4–5 visits, 6–16+ visits. By utilizing these categories, we were able to investigate SCD status and frequency of emergency room visits, which was not done in the earlier NHIS study [7]. This methodology is used for other variables included in our analyses as follows: number of doctor visits, reported health status, school absences, and emergency room visits. We cannot understate the significant issue, that not only are children with SCD confronted with healthcare barriers, but they also encounter academic challenges that could underscore the socioeconomic disparities.

### Strength and Limitations

Utilizing the NHIS dataset to characterize the impact of SCD for a population that is representative of individuals across the U.S., our study has several noted strengths, including but not limited to, sample size, multi-ethnicities representing the diversity of SCD patients, socioeconomic status, and insurance coverage in the U.S. Similar to our previous analysis for SCD and medical conditions, we examined SDoH in a U.S. based population that substantiates the generalizability of our findings for neurological outcomes [24].

The limitations of our study include, (1) selection bias—although it is highly unlikely, we cannot state that selection bias did not influence some of the associations in this secondary analysis. (2) Recall bias—because other data were not available to confirm the NHIS (e.g., medical records) recall bias [45] could be an issue for caregivers of children with SCD. Without confirmation of an SCD diagnosis, this analysis rests on uncertain data. An existing NHIS study of validity showed that under-reporting of medical conditions related to chronic disabilities by the child’s parent/guardian is likely to occur [44]. (3) SCD genotypes are not represented in the NHIS dataset, because SCD genotypes are associated with varying severities and frequencies of complications; some parents/guardians might not be able to differentiate between sickle cell anemia (SCA) and SCD [24].

## 5. Conclusions

In this representative sample, U.S. children with SCD have higher odds of neuro-developmental conditions influenced by SDoH (e.g., SES and healthcare access), with Black children shouldering a disproportionate burden. A similar pattern was identified in that Black children with SCD reported increased healthcare and special education services utilization in the past 12 months [24]. This work highlights significant issues that should be addressed by health providers and within the elementary and secondary school environment. This creates the urgency to address the health burden for Black children with SCD that has been exaggerated by barriers to accessing comprehensive healthcare due to SDoH. Studies that further assess and implement effective interventions to combat children’s health burden with SCD, neurocognitive impairment, and healthcare utilization are crucial to improving their quality of life and achieving health equity.

## Figures and Tables

**Figure 1 ijerph-20-06137-f001:**
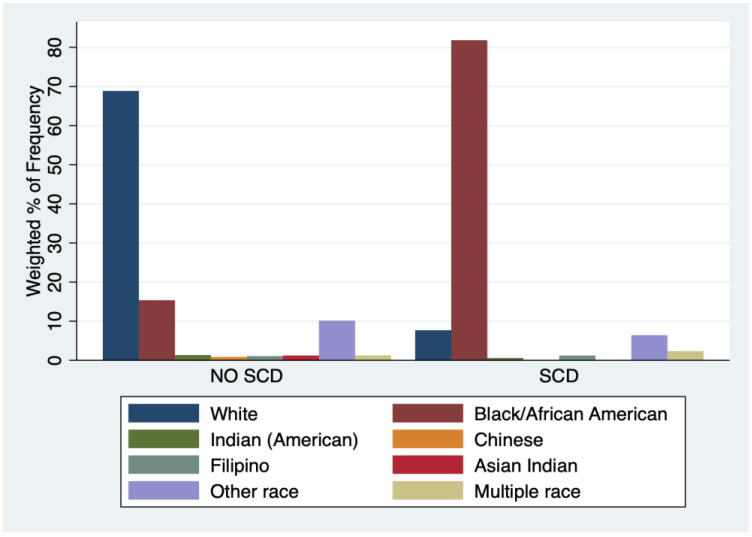
Sickle Cell Disease Status by Race/Ethnicity Data from NHIS.

**Figure 2 ijerph-20-06137-f002:**
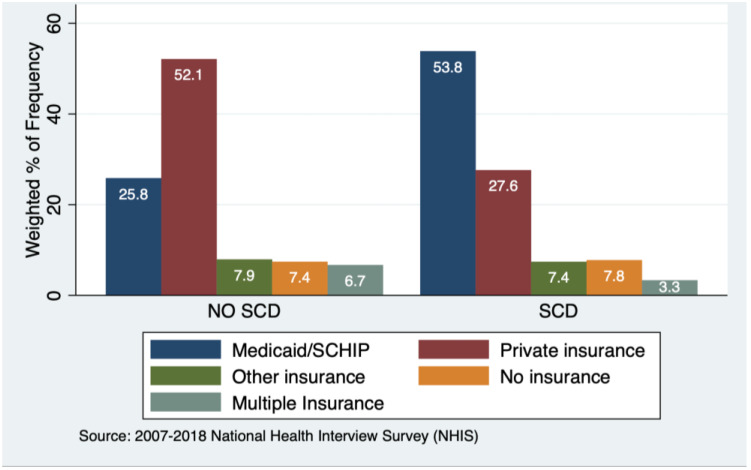
Insurance profile of respondents to the NHIS based on Sickle Cell Disease Status.

**Table 1 ijerph-20-06137-t001:** (**a**): Developmental conditions among all children by SCD status. (**b**): Health status and healthcare use for all children by sickle cell disease (SCD) status.

Table 1(a)						
	Children with SCD (*n* = 215)		Children without SCD (*n* = 133,266)		AOR for SCD vs. No SCD (95% CI)	*p* Value
**Developmental Conditions**	**Unweighted**	**Weighted %**	**Unweighted *n***	**Weighted %**		
ADD/ADHD	27	14.32	9754	8.34	0.8 (0.3, 2.0)	0.657
Learning Disability	28	17.08	8497	7.70	1.5 (0.6, 3.5)	0.350
Intellectual Disability	5	4.11	1230	0.99	4.1 (1.3, 13.3)	0.018
Trouble Hearing	10	5.28	2679	2.09	2.5 (1.0, 6.0)	0.039
Trouble Seeing	10	5.67	3718	2.80	2.0 (0.8, 4.6)	0.114
Other developmental delay	2	5.21	268	1.97	0.9 (0.1, 7.3)	0.947
**Table 1(b)**						
	**Children with SCD** **Unweighted** **(*n* = 215)**		**Children without SCD** **Unweighted** **(*n* = 133,266)**		**AOR for SCD vs. No SCD (95% CI)**	***p* Value**
**Health Impact**	**Unweighted *n***	**Weighted %**	**Unweighted *n***	**Weighted %**		
Limited in ability to crawl, walk, run, or play	22	12.70	2340	1.75	7.2 (3.9, 13.3)	<0.001
Needed special equipment	4	1.47	1566	1.16	1.3 (0.4, 4.4)	0.647
Took prescription medication for at least three months	78	36.44	17,308	12.95	3.8 (2.4, 6.2)	<0.001
**Reported health status fair or poor**						
Better	69	33.42	29,475	20.73	1.5 (1.0, 2.3)	0.035
Worse	10	4.63	1792	1.33	4.1 (1.8, 9.0)	0.001
About the same	135	61.95	101,648	77.95	0.5 (0.4, 0.8)	0.003
**Miss school because of illness or injury**						
0	29	24.85	28,969	30.12	0.4 (0.2, 0.8)	0.007
1–10	78	59.37	60,803	65.45	1.1 (0.7, 1.8)	0.723
11–20	14	8.82	3176	3.26	3.8 (1.8, 7.7)	<0.001
21–30	5	2.84	746	0.71	6.5 (2.4, 17.6)	<0.001
31–40	0	-	174	0.15	-	-
40+	5	4.11	328	0.31	14.0 (4.5, 43.1)	<0.001
**Healthcare and special education services use (past 12 months)**						
Saw a medical specialist	56	23.56	18,836	14.29	2.3 (1.5, 3.7)	<0.001
Saw mental health professional	22	9.82	9219	7.81	1.04 (0.5, 2.2)	0.911
Saw a therapist	22	13.20	8558	7.55	2.0 (1.0, 4.1)	0.040
Saw an optometrist, ophthalmologist, or eye doctor	57	31.20	31,873	26.89	1.4 (0.9, 2.3)	0.126
**How many times have you seen a doctor within 12 months?**						
none	9	3.06	12,651	9.19	0.2 (0.1, 0.6)	0.003
1–5	142	68.47	99,137	75.91	0.5 (0.4, 0.8)	0.004
6–9	28	14.72	11,842	8.82	2.6 (1.4, 4.8)	0.002
10–15	21	9.55	5239	3.86	3.5 (1.9, 6.4)	<0.001
16 or more	12	4.20	2921	2.22	2.2 (0.9, 5.2)	0.073
**Multiple visits to emergency room**						
none	125	55.32	106,956	81.24	0.4 (0.2, 0.6)	<0.001
1	34	15.75	16,969	12.50	1.0 (0.5, 1.7)	0.935
2–3	32	17.28	6869	5.01	2.9 (1.6, 5.2)	0.001
4–5	13	6.54	1089	0.79	4.8 (2.3, 10.2)	<0.001
6–16+	11	5.11	634	0.45	9.3 (3.6, 24.0)	<0.001
Had surgery or another surgical procedure	15	3.92	6286	4.76	1.0 (0.5, 2.0)	0.957
**Healthcare Barrier**						
Couldn’t use telephone	5	3.62	2330	1.74	1.4 (0.4, 5.5)	0.613
Couldn’t get an appointment	11	5.66	5882	4.45	1.1 (0.5, 2.5)	0.858
Waited too long at the doctor’s office	11	2.63	5457	3.95	0.6 (0.2, 1.3)	0.163
No transportation	12	4.42	2416	1.83	0.8 (0.3, 1.9)	0.646
Couldn’t afford prescription medicines	1	5.45	230	1.37	5.1 (0.9, 28.3)	0.065

**Table 2 ijerph-20-06137-t002:** (**a**): Developmental conditions among Black children by SCD status. (**b**): Health status and healthcare use for Black children by sickle cell disease (SCD) status.

Table 2(a)						
	Black Children with SCD Unweighted (*n* = 170)		Black Children without SCD Unweighted (*n* = 21,395)		AOR for SCD vs. No SCD (95% CI)	*p* Value
Developmental Conditions	Unweighted *n*	Weighted %	Unweighted *n*	Weighted %		
ADD/ADHD	23	14.05	1806	9.43	5 (0.2, 1.2)	0.117
Learning Disability	23	17.64	1563	8.96	1.3 (0.5, 3.1)	0.620
Intellectual Disability	2	2.49	239	1.25	2.4 (0.6, 9.9)	0.232
Trouble Hearing	8	3.87	524	2.56	1.6 (0.6, 4.1)	0.303
Trouble Seeing	10	6.93	698	3.64	2.3 (1.0, 5.4)	0.047
Other developmental delay	2	7.36	38	1.58	1.9 (0.2, 23.0)	0.618
**Table 2(b)**						
	**Black Children with SCD** **Unweighted** **(*n* = 170)**		**Black Children without SCD** **Unweighted** **(*n* = 21,395)**		**AOR for SCD vs. No SCD (95% CI)**	***p* Value**
**Health Impact**	**Unweighted *n***	**Weighted %**	**Unweighted *n***	**Weighted %**		
Limited in ability to crawl, walk, run, or play	21	13.63	399	1.83	7.8 (4.2, 14.4)	<0.001
Needed special equipment	3	1.45	263	1.23	1.3 (0.3, 5.5)	0.707
Took prescription medication for at least three months	69	38.90	3139	14.00	3.9 (2.3, 6.4)	<0.001
**Reported health status fair or poor**	**Unweighted *n***		**Unweighted *n***			
Better	58	36.73	5124	24.10	1.8 (1.2, 2.9)	0.008
Worse	7	3.74	258	1.35	3.1 (1.0, 9.2)	0.041
About the same	104	59.52	15,920	74.54	0.5 (0.3, 0.8)	0.001
**Miss school because of illness or injury (days)**	**Unweighted *n***		**Unweighted *n***			
0	24	26.78	5944	39.56	0.5 (0.2, 0.9)	0.028
1–10	64	55.81	8749	57.10	0.9 (0.5, 1.6)	0.758
11–20	13	9.38	407	2.42	4.1 (1.9, 9.0)	<0.001
21–30	5	3.28	102	0.55	8.4 (2.9, 24.3)	<0.001
31–40	0	-	17	0.01	-	-
40+	5	4.75	39	0.28	19.0 (5.5, 65.5)	<0.001
**Healthcare and special education services use (past 12 months)**	**Unweighted *n***		**Unweighted *n***			
Saw a medical specialist	47	23.52	2507	11.56	2.5 (1.5, 4.2)	0.001
Saw mental health professional	19	10.74	1417	7.38	1.1 (0.5, 2.3)	0.900
Saw a therapist	16	12.56	1258	6.83	2.0 (0.9, 4.1)	0.074
Saw an optometrist, ophthalmologist, or eye doctor	50	32.18	4878	25.37	1.5 (0.9, 2.5)	0.144
**How many times have you seen a doctor within 12 months)?**	**Unweighted *n***		**Unweighted *n***			
none	8	3.57	2022	9.25	0.2 (0.1, 0.7)	0.009
1–5	111	69.22	16,583	79.38	0.6 (0.4, 0.9)	0.019
6–9	22	13.30	1510	6.98	2.3 (1.2, 4.5)	0.010
10–15	17	9.44	628	2.92	3.7 (1.9, 7.3)	<0.001
16 or more	10	4.47	325	1.48	2.6 (1.0, 6.8)	0.059
**Multiple visits to emergency room**	**Unweighted *n***		**Unweighted *n***			
none	91	52.68	16,058	76.34	0.3 (0.2, 0.6)	<0.001
1	30	17.42	3173	14.66	1.1 (0.6, 1.9)	0.773
2–3	28	18.14	1572	7.15	2.8 (1.5, 5.4)	0.002
4–5	11	7.31	266	1.15	5.1 (2.3, 11.3)	<0.001
6–16+	10	4.45	148	0.69	7.3 (3.0, 17.5)	<0.001
Had surgery or another surgical procedure	13	3.82	824	3.89	1.0 (0.4, 2.0)	0.905
**Healthcare Barrier**	**Unweighted *n***		**Unweighted *n***			
Couldn’t use telephone	5	4.42	361	1.79	1.7 (0.4, 6.7)	0.431
Couldn’t get an appointment	8	6.14	964	4.58	1.2 (0.5, 3.0)	0.717
Waited too long at the doctor’s office	4	1.33	808	3.87	0.3 (0.1, 1.1)	0.064
No transportation	10	4.70	622	3.24	0.8 (0.3, 2.1)	0.632
Couldn’t afford prescription medicines	1	7.70	41	1.30	8.4 (2.4, 29.4)	0.001

## Data Availability

The data presented in this study are openly available US Centers for Disease Control and Prevention website [https://www.cdc.gov/nchs/nhis/data-questionnaires-documentation.htm] (accessed on 16 September 2022).

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
