# Peer review of "Analysis of the 2007–2018 National Health Interview Survey (NHIS): Examining Neurological Complications among Children with Sickle Cell Disease in the United States"

_ijerph, 2023, doi:10.3390/ijerph20126137_

Round 1

Reviewer 1 Report

Statistical significativity must be express in a more formal way.
p must be = to a number (i.e. p=0.028 - line 225; p = 0.012 - line 247, ...) or p< 0.001 if the result is much lower than 0.001

In the tables, some p values are bold and other no (line 214 also if significatant)

In the format shown by pdf, numbers in the tables are not clearly readble: example: 8.4(2.4,29.4). Better to use spaces and "-": 8.2 (2.4 - 29.4).

Author Response

Responses to Reviewers’ Critiques

May 18, 2023

Manuscript ID: ijerph-2296020

Title: Analysis of the 2007-2018 National Health Interview Survey (NHIS): Examining neurological complications among children with Sickle Cell Disease in the United States

We thank the editor and the reviewers for their constructive feedback and have addressed all comments in the manuscript. Furthermore, we have provided specific responses on how we have addressed the comments from the reviewers (please see edits in marked up version of the manuscript).

REVIEWER 1

Review 1, Comment 1. Statistical significatively must be express in a more formal way.
p must be = to a number (i.e. p=0.028 - line 225; p = 0.012 - line 247, ...) or p< 0.001 if the result is much lower than 0.001

Author Response: We thank this reviewer for the thorough review of the manuscript. We have revised the manuscript to address these comments in lines 225, and 247 and in the entire manuscript.

Review 1, Comment 2. In the tables, some p-values are bold and other no (line 214 also if significant)

Author Response: We have revised the manuscript to address these comments in lines 214, and in the entire manuscript.

Review 1, Comment 3. In the format shown by pdf, numbers in the tables are not clearly readable: example: 8.4(2.4,29.4). Better to use spaces and "-": 8.2 (2.4 - 29.4).

Author Response: We have revised the manuscript and reformatted the tables.

REVIEWER 2

Review 2, Comment 1. It is a big and interesting analysis of Sickle cell disease in USA.

As SC Disease is a multisystemic disease and Neurological complications are related also to the therapy and the molecular background.

Author Response: We thank the reviewer for this supportive comment. The clinical and research contributions to the sickle cell disease literature for the patients in the United States is significant.

Review 2, Comment 2. The therapy (Hydroxyurea, blood transfusion methods performed) could add to your research concerning the neurological profile. Neuroimaging is crucial and could be added if possible by the records of the patients.

Author Response: We thank this reviewer for this thoughtful suggestion. Unfortunately, the National Health Interview Survey (NHIS) does not have any medical records associated with the patient/care giver responses. This is a significant limitation of our study because it is difficult to confirm responses via medical records, we have noted this limitation in the discussion section.

Review 2, Comment 3.  Could you explain, SDOH, SES in few words for the readers.

Author Response: We thank this reviewer for this comment and have added a few sentences on SDoH and SES on pages 3-4 in the Introduction section of the manuscript to provide explanations and its relevance to sickle cell disease.

REVIEWER 3

Review 3, Comment 1. line 44: the incidence of SCD in US Blacks is 1/365 births but what is the national incidence? It is not specified. And if the incidence in your paper is lower or higher that in the national population, it must be discussed.

Author Response: We thank this reviewer for the thorough reading of the manuscript. It is difficult to find the national incidence for sickle cell disease in the US for Blacks because the US does not have a national registry for sickle cell disease nor do many states have state-level data. At most 10 states have state-level data on sickle cell disease births. At best, the prevalence data for sickle cell disease is a hypothetical guess based on calculated birthrates and not based on empirical national-level data collection.

Review 3, Comment 2. line 126: categories of duration for missing school: is it an annual rate as for the annual physician visits? it is not clear

Author Response: The duration of missing school is summarized in an annual rate.

Review 3, Comment 3. table 1 - line 168 and methodology: it is possible that the same child was assessed in different year of survey?

Author Response: We thank this reviewer for the thorough review of the manuscript. Yes, it is entirely probably that the same child was assessed every year of the survey especially for a specific and rare condition such as sickle cell disease.

Review 3, Comment 4. Figure 1 and 2 (line 188 - 189): these figures provide nothing more than the data in the tables (especially since they have already been published in the previous BMJ paper).

Author Response: Thank you for this insightful comment. We have removed figures 1 and 2 from the manuscript and have reference the BMJ paper.  

Review 3, Comment 5.  lines 288-293: Just because black children with sickle cell disease are more likely to utilize healthcare service doesn't mean this group is not receiving the necessary healthcare services. It is normal that with a chronic condition they have frequent visits and this should be compared to other patients with chronic diseases. 

Author Response: We thank the reviewer for the thorough reading of the manuscript and agree that in principle higher healthcare utilization does not equal inadequate care. However, when the totality of the data is examined including poor outcomes with the current existing literature which highlight higher stigmatization, racial discrimination, and other structural barriers that persons with sickle cell disease encounter (Lee T. et al.Public Health Rep. 2019;134(6):599-607) in additional to the SDoH, the NHIS data is one of many data points that demonstrate health inequities affecting persons with sickle cell disease exist. Other publications also demonstrate the lack of timely care for persons with sickle cell disease (Haywood C. et al. J Pain Symptom Manage. 2014;48(5):934-943).

Review 3, Comment 6.  I well understood the concept of the NHIS survey based on the sample child core questionnaire, however the questionnaire-related biases and the fact that you don't take account of medical record or specific testing should be discussed in the methodology or in the limitations of the study.

Author Response: We thank this reviewer for the thorough review of the manuscript. Within the Strength and Limitations section after the Discussion in the manuscript we highlight three significant limitations of the NHIS that include 1) selection bias, 2) recall bias by responders that cannot be confirmed by medical records, and finally 3) lack of sickle cell disease (SCD) genotypes that can be helpful in differentiating between sickle cell anemia (SCA) and SCD. The questionnaire asked about sickle cell anemia but many care givers might not differentiate between SCD and SCA. Inclusion of medical records, and/or genotypic data within the NHIS would have produced a rigorous dataset and subsequent manuscript.  

Author Response

(The authors gave the same response as above.)

Reviewer 3 Report

Dear Authors,

Congratulations for your work.

First, some minor issues : 

> line 44 : the incidence of SCD in US Blacks is 1/365 births but what is the national incidence ? It is not specify. And if the incidence in your paper is lower or higher that in the national population, it must be discussed.

> line 126 : categories of duration for missing school : is it an annual rate as for the annual physician visits? it is not clear 

> table 1 - line 168 and methodology : it is possible that the same child was assessed in different year of survey?

> Figure 1 and 2 (line 188 - 189) : these figures provide nothing more than the data in the tables (especially since they have already been published in the previous BMJ paper)

The others comments relate to some interpretation and the discussion of the biases : 

> lines 288-293 : Just because black children with SCD are more likely to utilize healthcare service doesn't mean this group is not receiving the necessary healthcare services. It is normal that with a chronic condition they have frequent visits and this should be compared to other patients with  chronic diseases. 

> I well understood the concept of the NHIS survey based on the sample child core questionnaire, however the questionnaire-related biases and the fact that you don't take account of medical record or specific testing should be discussed in the methodology or in the limitations of the study.

Author Response

(The authors gave the same response as above.)
